# Recent Advances in Molecular Improvement for Potato Tuber Traits

**DOI:** 10.3390/ijms23179982

**Published:** 2022-09-01

**Authors:** Daraz Ahmad, Zhongwei Zhang, Haroon Rasheed, Xiaoyong Xu, Jinsong Bao

**Affiliations:** 1Institute of Nuclear Agricultural Sciences, College of Agriculture and Biotechnology, Zhejiang University, Xihu District, Hangzhou 310058, China; 2Yazhou Bay Seed Laboratory, Yazhou Bay Science and Technology City, Yazhou District, Sanya 572025, China; 3Hainan Institute, Zhejiang University, Yazhou Bay Science and Technology City, Yazhou District, Sanya 572025, China

**Keywords:** QTL mapping, genome selection, RNAi, CRISPR, plant breeding tools

## Abstract

Potato is an important crop due to its nutritional value and high yield potential. Improving the quality and quantity of tubers remains one of the most important breeding objectives. Genetic mapping helps to identify suitable markers for use in the molecular breeding, and combined with transgenic approaches provides an efficient way for gaining desirable traits. The advanced plant breeding tools and molecular techniques, e.g., TALENS, CRISPR-Cas9, RNAi, and cisgenesis, have been successfully used to improve the yield and nutritional value of potatoes in an increasing world population scenario. The emerging methods like genome editing tools can avoid incorporating transgene to keep the food more secure. Multiple success cases have been documented in genome editing literature. Recent advances in potato breeding and transgenic approaches to improve tuber quality and quantity have been summarized in this review.

## 1. Introduction

Potato (*Solanum tuberosum* L.) is the most important non-grain food crop concerning its global consumption [1]. It was firstly domesticated in Perú around 10,000 years ago. The potato will become an important crop consumed as staple food in the future due to increasing food security issues, since the global population will rise to 9.7 billion by 2050 [2,3,4].

Potato has high nutritional values due to the presence of several essential phytochemicals with significant health benefits [5]. Various phytonutrients (carotenoids, anthocyanins, phenolics, and flavonoids), mineral elements (potassium, sodium, magnesium, copper, iron, and zinc), and vitamins (vitamin B1, B6, B9, C, E) are the major components of potatoes [6,7]. Due to their antioxidant properties, these phytonutrients are essential for maintaining human health [5,8].

The main motive for improving this crop is to improve the processing tuber quality with a higher yield. Yield of potatoes is determined by proper genotypes and growing environments, and is significantly correlated with the number of sprouts, plant height, shoot diameter, and leaves length [9]. The market values of table potatoes depend on the visual features of cultivars, i.e., depth of eyes, flesh color, tuber shape, and cooking suitability. The chemical composition of the tuber impacts the eating and appearance quality of the potato products, for example, starch contents impact cooked product texture, and sugar content directly affects fried product color [10].

Research on the genome sequencing of cultivars and their wild relatives have revealed more significant genetic variations and signs of selection of theinteresting traits via genes. Potato shows higher heterozygosity, tetrasomic polyploidity owing to its four homolog chromosomes, rendering challenges to alter its genome [11]. Scrutinizing genes with functional and molecular genetics approaches will also meet key hindrances [12]. Thus, combining the knowledge of functional genomics and basic plant breeding techniques with advanced molecular methods is crucial to improve agronomic traits. Though many review articles on potato breeding and transgenic potato have been written [10,11,12,13,14], there is a lack of a comprehensive review focusing on a bridge between potato breeding and advanced molecular techniques for improving the quality and yield of potato tubers. This review will provide a new direction for potato breeders to use a combined approach of advanced breeding and molecular techniques.

## 2. Quantitative Trait Loci (QTL) Mapping

Genetic modification does not rely merely on making crosses of donor species with the recipient cultivar of potato. Instead, molecular techniques offer an alternative approach, and reducing the chances of linkage drag associated with conventional methods. These methods can incorporate entire genes, gene groups, or even whole chromosomes [14]. Genotyping by next-generation sequencing tools can find the allelic variation in the genomics era (Figure 1). Various breeding tools such as QTL analysis, whole-genome sequencing, and marker-trait associations are crucial to understand the genetic differences between breeding clones, progeny crosses, and wild species (Figure 1). Wild germplasm shows a smaller genetic distance in self-incompatible germplasm than accessions of similar species in the germplasm [15].

### 2.1. QTL Mapping for Tuber Quality

Indirect manipulation and advances in QTL mapping have eased the way of crop tuber quantity and quality improvement. This article mainly describes QTL mapping and advanced molecular techniques based on the genetic improvement of potato tubers. Some major QTLs related to the potato tuber quality and quantity and their position regarding chromosomes are shown in Table 1.

Biosynthesis of “leaf starch” or “transient starch” usually occurs during the daytime [24]. In contrast, the breakdown of sucrose and its conversion to storage starch occur during the nighttime [24,25]. This conversion involves complexed procedures encoded by various gene sets located at 123 loci on I-XII chromosomes [26]. They also control tuber starch contents, sugar contents, and sugar-starch interconversion [26].

Starch granules in potato tubers are always more significant in size than the granules synthesized in the leaves [27,28]. For the size of starch granules, two QTLs viz. SGS02-8 and SGS03-8 have been documented on chromosome VIII [16]. Also, the structure and composition of storage starch are different from leaf starch [24,29]. Twelve QTLs related to tuber starch contents have been discovered (Table 1) [17]. Similarly, Li et al. [18] also reported other loci for starch contents (Table 1). Around 77 genomic loci encoding enzymes have been found associated with starch metabolism [30]. Various enzyme isoforms involved in starch metabolism also have been reported: 6 sub-units of ADP-glucose pyrophosphorylase large subunits; 5 different types of alpha-amylase; 4 different types of alpha-glucan phosphorylase; 2 different kinds of ATP-ADP antiporters; 10 different types of Beta-amylases; 4 different kinds of branching enzymes; disproportionating enzymes; glucan water dikinase (GWD); glucose transporters; glucose-6-phosphate translocators; granule bound starch synthases and many others [30]. The starch synthesis starts from bioconversion to adenosine diphosphate-glucose from sucrose, the primer substrate of starch biosynthesis, which are employed to synthesize the amylopectin and amylose. Five chromosomal regions on chromosome 2, 3, 5, 7, and 10 were linked with amylose content [16].

The distinctiveness of potato starch is that it has high level of starch-bound phosphate, conferring it new properties such as high peak viscosity and gel-forming capacity. A total of 17 loci was identified significantly for the phosphorus content in potato starch, 14 of which are assigned to 8 genomic regions on chromosomes 1, 4, 5, 7, 8, 10, and 11, and most of the SNPs identified belong to protein coding regions [31].

Allelic variation significantly affects the starch phosphorylation process, and the genes encoding GWD, starch branching enzymes I and II, and starch synthase III are associated with this process [32]. These allelic variants can be served as genetic markers for starch phosphorylation [32].

The important loci related to tuber starch contents were firstly documented by Li et al. [20]. QTLs related to starch contents were AmyZ (M79328), SssI (Y10416), StpL (X73684), GbssI (X52417), StpH, located on chromosomes IV, III, V, VIII and IX, respectively [20]. Another genetic mapping study has revealed 12 different loci regarding tuber starch contents viz. pPt-539763, pPt-535988, toPt-438845, pPt-533878, pPt-471789, pPt-538127, toPt-437014–pPt-538033, toPt-440651, and pPt-656237, on seven chromosomes: I, II, III, VIII, X, XI, and XII [17]. Average starch yield and chips quality are associated with Stp23-8b and Pain1-8c or Pain1prom-d/e [20,33].

Protein content is another importance quality trait of the potato industry. Genetic mapping analysis revealed potential QTLs on chromosomes 2, 3, 5 and 9, and further cofactor QTL analysis identified two masked QTLs on chromosomes 1 and 5 [34].

### 2.2. QTL Mapping for Tubers Storage Duration

Accumulating reducing sugars in cold-stored potatoes, called cold-induced sweetening (CIS), is a severe problem that causes unacceptable color changes and acrylamide formation in fried products. CIS has been found to be associated with various enzymes, such as starch synthases, adenosine diphosphate glucose pyrophosphorylase, branching enzymes, debranching enzymes, β-amylases, amylase inhibitors, and starch phosphorylases for degradation of starch contents [35]. The CIS has been related to the putative Kunitz-type tuber invertase inhibitor (KT-InvInh) position on the StKI locus at chromosome III [36].

The low sugar and fructose contents of tubers play pivotal roles in the processing quality of the potato. The organization of invertase genes (*InvGE* and *InvGF*) at the Inv ap -b locus has already been documented on chromosome IX [37]. Later, *InvGE*-*A 85 (A 86)*, *InvCD141*-*G 765*, *InvCD141_T 543 (A 280*, *T 288*, *T 339*, *A 630*, *C 1030*, *G 1031*, *T 1096) InvGE*-*G 95 (G 106)*, *Pain1*- *A 718 (C 552) 2*, and *Pain1*- *A 1544*, *Pain1*-*T 741* on chromosomes III, X and IX were found to be associated with tuber starch contents and its processing quality [19]. The QTLs of CIS and reconditioning (REC; a process of reducing sugars into starch by storing the cold-stored potatoes) were identified on chromosomes V, VI, and VII of the CIS-susceptible and chromosomes V and XI in the CIS-resistant parents. Two functional genes, a starch hydrolysis gene *GWD*, were located with QTL as REC_B_05-1, and a starch synthesis gene *AGPS2* has also been mapped in QTL CIS_E_07-1 [21].

The QTLs for tuber dormancy and sprouting traits were found on chromosomes 2, 3, and 7, which are good candidates for marker-assisted breeding [38].

### 2.3. QTL Mapping for Tuber Morphology

Tuber shape and color are very important quality traits regarding its market life or selling because consumers pay more attention to the shape of potato tubers. For the tuber shape, two loci, SNP Solcap_snp_c2_25510 (Alpha virus core protein family) and Solcap_snp_c2_25485 (ribosomal protein S6 kinase) on chromosome X were found significantly associated with the trait in 2016 [22], while another locus, SNP Solcap_snp_c2_34875 (serine/threonine-protein kinase) on chromosome number IV was found to be significant in 2017. For tuber skin color, a total of 12 QTLs, three on chromosome 1; one on chromosome 5; two on chromosome 7; three on chromosome 8; one on chromosome 10; one on chromosome 11, and one on chromosome 12 were detected (Table 1) [22]. Significant QTLs for tuber shape were detected on chromosomes 4, 7, and 10, with heritability estimates ranging from 0.09 to 0.36 [23]. For each grain shape trait (eye depth, tuber shape, regularity of tuber shape, mean tuber weight), four to seven QTLs were determined except for tuber flesh color, which was controlled by a major QTL on chromosome III, accounting for 76.8% of trait variance [39]. Moreover, a minor QTL was localized on chromosome II for flesh color, [39]. In a gynogenic dihaploid (2n = 2x = 24) population, three QTLs for tuber shape were identified on chromosomes 6, 10, and 11 [40].

Likewise, certain other QTLs linked with tuber morphology and pigmentation have been documented. For instance, the *R2R3*-*MYB* loci for tuber flesh color have been revealed on chromosome 10. *Dihydroflavonol 4*-*reductase* and *flavonoid 3**′*,*5**′*-*hydroxylase* are associated with color, whose loci have been reported on chromosomes 2 and 11 [41]. Similarly, the loci of *Anthocyanin 1* associated with pigment flesh intensity was found on chromosomes 9 and 10, and that of *β*-*carotene hydroxylase* for yellow flesh was detected on chromosome 3 [41]. These mapping researches provide a better understanding of the genetics of various desirable characters and suitable diagnostic DNA markers for indirect marker aided selection (MAS) of target traits in potato breeding.

## 3. Genome Selection (GS)

In conventional breeding techniques, for several generations, recurrent phenotypic selection can be exploited [42]. Decreasing population size and increasing genotypes number under evaluation are the typical selection challenges in the conventional method. The breeder selects parents based on phenotypic characters for pair-wise crossing [43]. Generally, several high-yield and stress-resistant parents will be screened over several clonal generations in several years [44]. The environment significantly influences the expression of multiple traits in potatoes, such as yield, tuber size, and processing quality. Almost 40 traits are considered to be important for the cultivar development of potato, which are divided into yield, tuber quality, and stress tolerance attributes [45]. Knowledge of the gene × environment interactions will help the identification of superior parents, screening, and selection methods for superior genotypes [46]. Figure 1 shows a schematic flow of various steps involved in genome-assisted breeding.

MAS is a great tool for assisting plant breeding, but many agronomic traits are complicated and controlled by several loci. An improved and new breeding approach has been developed, which is genomic selection (GS). GS helps estimate plants’ genetic success based upon all molecular markers’ information simultaneously instead of a few ones like in MAS [47]. Increased accuracy of high-throughput genotyping techniques, decreased cost of sequencing techniques, and marker selection have reinforced the value of GS [48]. GS can predict the genetic value/breeding value. There are two populations: the training population that is genotyped and phenotyped, and the testing population that is genotyped but not phenotyped. GS is used to predict the phenotype of testing population for some selection cycles. Various statistical models are used to predict the traits of interest, such as Kinship GAUSS, PR-BLUP, Bayesian LASSO, Bayes B, and Bayes Cp [49]. For starch contents, the cross-prediction validation correlation of 0.56 in the training panel but the correlation was around 0.30 in the test panel derived from the tetraploid mapping population of a breeding program in Denmark [50]. The prediction accuracy of starch contents has improved by 8% in tetraploid German breeding clones [51]. Correlations of cross-prediction validation regarding tuber dry matter or specific gravity were 0.75–0.83 and 0.37–0.71 among different populations in Denmark [52,53]. In contrast, average cross-validation of the dry matter in European cultivars was 0.65 (ranging from 0.54 to 0.68) [54]. For chipping quality, prediction accuracy ranged from 0.4 to 0.45, as depicted from the pedigree depth of unselected US populations [52]. When tetraploid mapping populations were used in the Denmark breeding program, cross-prediction validation correlations ranged from 0.39 to 0.79, while the correlations declined across the populations, ranging from 0.28 to 0.48 [53]. The results gave a 0.73 cross-prediction validation correlation for the training panel, while in the test panel, 0.42 and 0.43 correlation values from mapping populations of tetraploid potato breeding programs in the rest of Europe [50]. For tuber yield and yield components, an additive genetic variance of SNPs captured 45% of the total genetic variance; the prediction accuracy was calculated in the range of 0.06 to 0.63 in tetraploid breeding unselected US F_1_ population [52]. In cultivated German clones of breeding potato, about an 8% increase was obtained in the prediction model accuracy using both additive and dominance effects [51]. Averages of 0.37, 0.32, and 0.17 for total yield, size, and number were revealed by cross-validation of several models, respectively [54]. Sallam et al. [55] indicated that if only the prediction accuracy exceeds 0.25, GS should surpass phenotypic selection in gain per unit time. Although the lowest average prediction correlation in some GS studies was around 0.30, GS still has potential to improve breeding efficiency in tetraploid potato for less cost per unit time.

Genome design breeding of hybrid potato has been carried out to exploit the heterosis in diploid potatoes with high homozygosity, enabling the transformation of potato breeding from a slow, non-accumulative mode into a fast iterative one [56]. The resultant hybrids showed strong heterosis [56]. For successful hybrid potato breeding, more inbred lines of high homozygosity are essential. Huang et al. [57] assembled 44 high-quality pan-genomes from 24 wild and 20 cultivated accessions that are representative of *Solanum* section *Petota*, the tuber-bearing clade, as well as 2 genomes from the neighbouring section, *Etuberosum*. The identification of 561,433 high-confidence structural variants and construction of a map of large inversions provides critical guidance for improving inbred lines and precluding potential linkage drag, which will accelerate hybrid potato breeding [57].

## 4. Transgenic Breeding

When gene identification tools combined with advanced biotechnological tools such as genome editing, incorporating the genes into the breeding program becomes fast [58,59,60,61,62]. Though there are many approaches to making transgenics, only four main approaches will be discussed here:(1)Conventional transgenic approaches of potato breeding include incorporation of a transgene via Agrobacterium-mediated transformation or any other vector for stable expression of a gene;(2)RNA interference (RNAi)-mediated transgenics is made to decrease the expression of undesirable traits by adding sense and antisense of the target gene with an intronic sequence. When this cassette is introduced into the plant genome, the target gene expression is significantly decreased;(3)Transcription Activator-Like Effector Nucleases (TALENs) mediated genome editing, which is utilized for generating non-GMO gene modification as well;(4)Clustered Regularly Interspaced Short Palindromic Nucleases (CRISPR)-associated (Cas) system-mediated genome-editing in potato plants. For this approach, a double-strand break is produced in the undesirable gene coding for an undesirable character. After a successful interruption, mutations are produced as required.

### 4.1. Conventional Potato Transgenics

#### 4.1.1. Transgenics to Improve Tuber Quality and Yield

An overview of the development of transgenic potato through agrobacterium-mediated transformation is given in Figure 2. Few examples in the modification of genes in potatoes via different transgenic engineering techniques are given in Table 2.

Starch yield has been increased by modifying potato source and sink capacities. This purpose was achieved by increasing source capacity via overexpression of mesophyll-specific pyrophosphatases or by producing antisense expression of ADP-glucose pyrophosphorylase in potato leaves. Re-routing of photoassimilates was carried out to deploy both approaches, thus making use of sink organs by consuming the leaf starch. Instantaneous enhancement in sink capacity was carried out by increasing the expression of two plastidic metabolite translocators, i.e., an adenylate translocator in potato tubers and a glucose 6-phosphate/phosphate translocator. Using this ‘pull’ approach, an increase in potato starch contents and starch yield have also been documented when sink strength is increased. In the recent biotechnological approaches, source and sink capacities were successfully enhanced by combining the “push” and “pull” approaches using two different attempts. This method led to two-fold increase in the starch yield in tubers. This successful approach can also be applied to other crop plants in the future [63]. The foreign sucrose-phosphate synthase gene has been successfully introduced in potatoes to improve the supply of photosynthate from leaves (source) down to the tubers (sink), resulting in better quantity and quality of potato tubers [65]. The agrobacterium auxin biosynthesis gene has also been successfully inserted in potato lines to enhance indole acetic acid contents in tubers and tuber formation [66]. Insertion of purple acid phosphatase 2 of Arabidopsis (AtPAP2) in potato lines resulted in increased tuber yield and tuber starch content [64]. Tuber size has also been improved in potato lines by inserting Arabidopsis *jasmonic acid carboxyl methyltransferase gene* [93].

Production of amino acids using genetic modified potato crops is particularly important for improving human nourishment [67]. Functional genomics studies conducted on the biosynthesis of amino acids have revealed that sucrose supply directly affects the increase in amino acids at the cell level and is fully controlled at transcription level, thus mediating the biosynthesis of amino acids in potatoes [94]. The methionine contents have been enhanced by increasing the expression of *CgSΔ90* in transgenic potato lines [95]. Cytidine base editor (CBE) was used to amino acid alterations, which led to a loss of function locus after substitutions of the KTGGL-encoding locus [96]. Scientists have successfully developed engineered potatoes inserted with the amaranth seed albumin (*AmA1*) gene or sunflower albumin gene increasing total methionine level 5–7 folds in tubers [67]. Different approaches have also been used to enhance methionine in potato plants by molecular breeders and biotechnologists [97,98,99]. Cysteine biosynthesis is associated with O-acetyl-L-serine produced by serine *acetyltransferase* (SAT). The SAT-encoding gene *cysE* was transformed in the white lady cultivar of potato, which significantly enhanced the transcript level of glutathione and cysteine in tubers to 1.5-fold higher on average than in the control plants [68].

Vitamins are pivotal in maintaining human health by mediating the metabolic system and bringing the metabolic processes associated with the energy in the tissues of living organisms obtained from food or other sources [100]. Many studies have been conducted on developing genetically modified potato tubers with increased vitamins. Vitamin A is functional in developing visual pigments, rod and cone cells inside the eye’s retina, synthesized from the precursor ß-carotene from potato. In potato, major carotenoids including antheraxanthin, lutein, xanthophyll, esters, and violaxanthin have been identified [101]. Researchers have successfully increased carotenoids to several folds in transgenic potatoes [70,102]. Research works have also revealed that the insertion of genes associated with the pathway of carotenoid biosynthesis along with the cauliflower orange (*Or*) gene in the potato genome increased the tuber astaxanthin contents successfully [69,71]. For successful commercial production of valuable ketocarotenoids in potato, the 3, 3′ β-*hydroxylase* (*crtZ*) and the 4, 4′ β-*oxygenase* (*crtW*) genes along with a suitable promoter were successfully inserted in the potato genome, which resulted in increased production of cellular carotenoids to form a wide array of ketolated and hydroxylated derivatives [103]. The *crtB* gene was successfully engineered in the potato genome using Agrobacterium to increase the carotenoid contents 4–7 times in *Solanum tuberosum* [104]. An algal *bkt1* gene, encoding a β-ketolase, has also been successfully transformed into the potato genome to enhance the levels of ketocarotenoid in potato tubers [71]. Similarly, the potato has been transformed by adding the *crtO* gene to successfully produce ketocarotenoids like astaxanthin (3,3′-dihydroxy 4,4′-diketo-β-carotene) [70].

Researchers have significantly enhanced potatoes’ ascorbic acid (AsA, vitamin C) contents by increasing the production of recycling ascorbate by inserting the dehydroascorbate reductase (*DHAR*) gene [74].

Vitamin E (α-tocopherol), a potential antioxidant, is biosynthesized by photosynthetic organisms and is crucial for human health, but it should be consumed at the sub-optimal level. Scientists have successfully overexpressed vitamin E in potato tuber by inserting with two vitamin-E biosynthetic genes, *homogentisate phytyl transferase* (*At*-*HPT*) and *p*- *hydroxyphenylpyruvate dioxygenase* (*At*-*HPPD*) [75]. Vitamin B6, an essential metabolite, is mandatory for living organisms because it is a cofactor in various biochemical reactions in the body. It is documented as a potential antioxidant molecule that regulates the protein expression involved in the scavenging of cellular reactive oxygen species. The *PDXII* gene was successfully inserted in potato lines to enhance the production of vitamin B6 [76].

#### 4.1.2. Transgenics to Improve Tuber Storage Duration

Successful engineering of novel RING finger gene *SbRFP1* in potato plants inhibited invertase and β-amylase activity. The activity of these enzymes decreased the degradation of sucrose and starch and thus decelerated the accumulation of reducing sugars in tubers [77].

Since the activity of vacuolar invertase (StvacINV1) is importantly involved in the CIS process, the invertase inhibitors can play roles in the inhibition of StvacINV1 activity. Scientists have successfully decreased the activity of StvacINV1 by adding two putative inhibitors (*StInvInh2A* and *StInvInh2B*) to the potato genome, which resulted in reduced accumulation of reducing sugars and acrylamide in cold-stored tubers and thus the CIS [78,79]. Sugar metabolism was significantly modified by inserting bacterial genes encoding phosphofructokinase to decrease CIS, and this addition did not hinder any growth process of the plant [80].

#### 4.1.3. Transgenics to Improve Tuber Morphology

Better tuber color and enhanced anthocyanin contents was successfully achieved by inserting a potato *UDP*-*glucose: flavonoid*-*3*-*O*-*glucosyltransferase* (*3GT*) gene in potato cultivar Désirée plants via agrobacterium-mediated transformation [105]. Similarly, the *R2R3 MYB* gene was added to the potato genome to increase the accumulation of pigments in the potato line [106].

### 4.2. RNAi for Potato Improvement

Since its discovery, RNAi technology has gained popularity in plant science [107]. Gene silencing at the post-transcription level has gained optimal results, which can be further applied in crop improvement.

#### 4.2.1. RNAi Transgenics to Improve Tuber Quality

The RNAi plant expression vectors including sense and antisense fragments of the *soluble starch synthase* (*SSIII*) gene was constructed and transformed into potato cultivars using Agrobacterium-mediated transformation method [81]. The transgenic potatoes produced different starch granules. The amylose content of starch was increased by 2.68–29.05%, while the amylopectin to amylose ratio and the phosphorus contents in the starch was significantly reduced compared to the control plants. So, it has been established that RNAi-mediated plants can be used for starch improvement in potato plants [81].

#### 4.2.2. RNAi Transgenics to Improve Tuber Storage Duration

The vacuolar invertase gene (*VInv*) was downregulated to more than 90%, reducing the sugars during cold storage by ~93% in the RNAi-mediated transgenic lines. Potato chips made with these transgenics were light in color because of the significantly low acrylamide content [84]. RNAi-mediated transgenics with the Désirée cultivar have also been produced to target the vacuolar invertase (*VInv*) gene, which converts starch into reducing sugars. The results were quite promising, like conventional transgenics, thus RNAi can be successfully used to reduce CIS in engineered potato tubers [83,84,85].

The activity of sucrose-phosphatase (SPP) was decreased in transgenic potatoes by RNAi, resulting in the decreased expression of vacuolar invertase upon cold treatment and thus decreased the CIS [86].

Acid invertases are key drivers among various enzymes, which playing an imperative role in sucrose to reducing sugar conversion. An average of 69.8% reduction in the expression of acid invertase in RNAi-mediated transgenic lines, decreased the CIS considerably. Compared to well-inhibited antisense *invertase* transgenic plants, RNAi mediated downregulation significant changed the invertase activity endogenously, showing its post-transcriptional gene silencing potential strategy in amelioration of cold sweetening in potato storage [108].

### 4.3. Genome Editing Technologies

#### 4.3.1. TALENs Mediated Potato Modifications

TALENs have been tested and proven functional in several studies of potato genome editing [109]. The *acetolactate synthesis* (*ALS*) gene encodes acetohydroxy acid synthase which catalyzes the first step in the synthesis of branched amino acid and is sensitive to a number of herbicides such as imazamox [110]. By using the TALEN system, the mutated *ALS* gene was coupled for the targeted integration of foreign genes into the host potato [111]. TALENs technology successfully improved tuber potato by vacuolar invertase activity interruption, which reduced CIS and reduced sugars [89]. Starch alterations were made by Kusano et al. [87], who designed the “Emerald-Gateway TALEN system”, a unique delivery system, and targeted *granule*-*bound starch synthase* (*GBSS*) gene in the host for site-specific mutation. It has its role in starch biosynthesis in granulation, thus modifying starch quality.

#### 4.3.2. CRISPR-Cas Mediated Potato Modifications

The overview of CRISPR-Cas9 mediated gene editing in an organism is depicted in Figure 3. The phytoene desaturase (PDS) has an imperative role in the carotenoid biosynthesis pathway, and caused the depigmentation due to the knockdown of PDS in the edited plants. Different sgRNAs have been utilized to decrease the activity of phytoene desaturase (PDS) by targeting its different sites. However, the unpaired nucleotides of target DNA can stimulate or decrease the activity of the Cas9-sgRNA complex in vitro according to the mismatch position [112].

The *GBSS* gene is considered to have a higher efficiency at the protoplast level. Replacing the Arabidopsis U6 promoter with endogenous potato *U6* promoter, resulting in 35% editing of the allelic genes in the ex-plants. However, a significant reduction in laborious cell culturing, regeneration of ex-plants, and tedious screening procedures of potatoes can be avoided with this tool [113]. Potato protoplasts were transfected with Ribonucleoprotein complexes (RNPs), consisting of sgRNAs and Cas9 nuclease. It resulted in mutations happened in around 68% of regenerated plants with at least one allele of the target gene. Successful mutations in the *StPPO2* gene resulted in a 69% reduction in the PPO activity in tubers and a 73% reduction in enzymatic browning compared to the control. Concludingly, a significant decrease in CIS can be made by CRISPR/Cas9 mediated genome editing [91].

The *StGBSS* gene was targeted via CRISPR-associated nuclease 9/guide-RNA (Cas9/gRNA), resulting in 25% mutagenesis in edited plants for GBSS [104]. Consequently, no enzyme activity related to carotenoids was observed due to the successful silencing of the *StGBSS* gene [112]. CRISPR-Cas9 has successfully altered the amylose-amylopectin ratio of potato starch. Gene encoding for GBSS was targeted from different sites, resulting in 67% mutant alleles in the edited plants. Full knockout of *GBSS* resulted in four allele-mutated lines of potato. Knockout of amylose alleles led to an enhanced amount of amylopectin contents [90].

The steroidal glycoalkaloids (SGAs) add a bitter taste to the potato tubers and have shown toxicity to various organisms. Fortunately, scientists have successfully decreased the SGA contents in the potato tubers. Nine candidate guide RNAs (gRNA) were prepared to target *St16DOX* in order to manipulate the characteristics of potato tubers according to the customer preference [92]. The gene *StSSR2* was targeted, which decreased the level of SGAs in edited plants by 44% in leaves and 66% in tubers [114].

### 4.4. Production of Non-GMOs Using Advanced Biotechnological Techniques

In this modern age of technology, there are several ethical, scientific, and public concerns about releasing genetically modified organisms (GMOs) as cultivars. The organism genetic components are better undisturbed for food security purpose except the acceptable genome modifications. TALENs and CRISPR-Cas9 have the potential to realize the objectives that genetically new but transgene-free plants can be obtained in T_2_ mutant lines after segregating generations from T_0_ [115].

## 5. Conclusions

QTL mapping has provided the molecular markers for marker assisted selection for potato yield and quality improvement. Genomic selection uses genome wide markers and facilitates rapid selection of potato breeding lines with superior yield and quality performance and increase the accuracy of selections. Many improvements in potato yield and nutrition levels regarding starch contents, protein contents, and vitamins, among other, have been achieved by conventional transgenic engineering and current genome editing technologies, e.g., TALENS and CRISPR-Cas9. Future researches can be focused on production of various vaccines in potatoes, which can easily be used to fight global pandemics like coronavirus, hepatitis, ebolavirus. Since population rise and arable land decline have always threatened our world, these new technologies are expected to increase yield and nutrition in potatoes that provide insights to fighting potential hunger in the growing world.

## Figures and Tables

**Figure 1 ijms-23-09982-f001:**
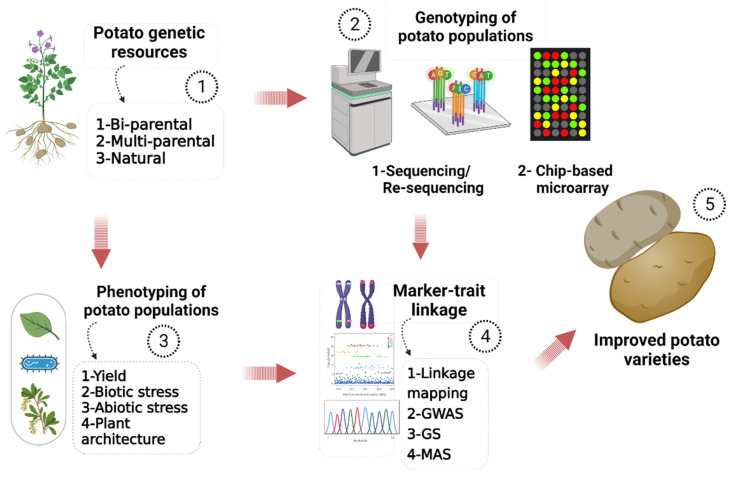
A schematic flow diagram of genome-assisted breeding. ① A core collection of *S. tuberosum* germplasm exhibits the trait of interest and wide genetic diversity. ②/③ After genetic resource selection, a trait-based phenotyping assessment is performed, followed by genotyping of selected populations. ④ Genotype-phenotype linkage is detected through various techniques. ⑤ The breeding program results in the development of improved potato varieties.

**Figure 2 ijms-23-09982-f002:**
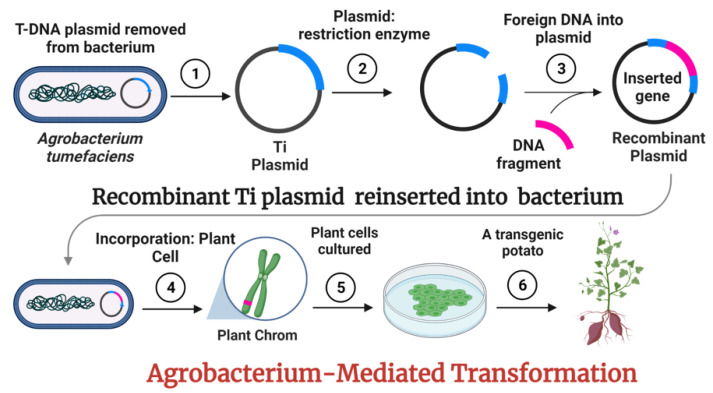
An overview of agrobacterium-mediated gene transformation in plants. ① Ti-Plasmid isolation from parent cell (*Agrobacterium tumefaciens*). ② Digestion of the plasmid with specific endonucleases enzymes. ③ Ligation of foreign DNA into the plasmid. ④ Insertion of recombinant-plasmid into the bacterium and its incorporation into plant’s cell nuclear DNA after infection. ⑤ Growth of transgenic plant tissues artificially. ⑥ Development of transgenic potato grown in a field.

**Figure 3 ijms-23-09982-f003:**
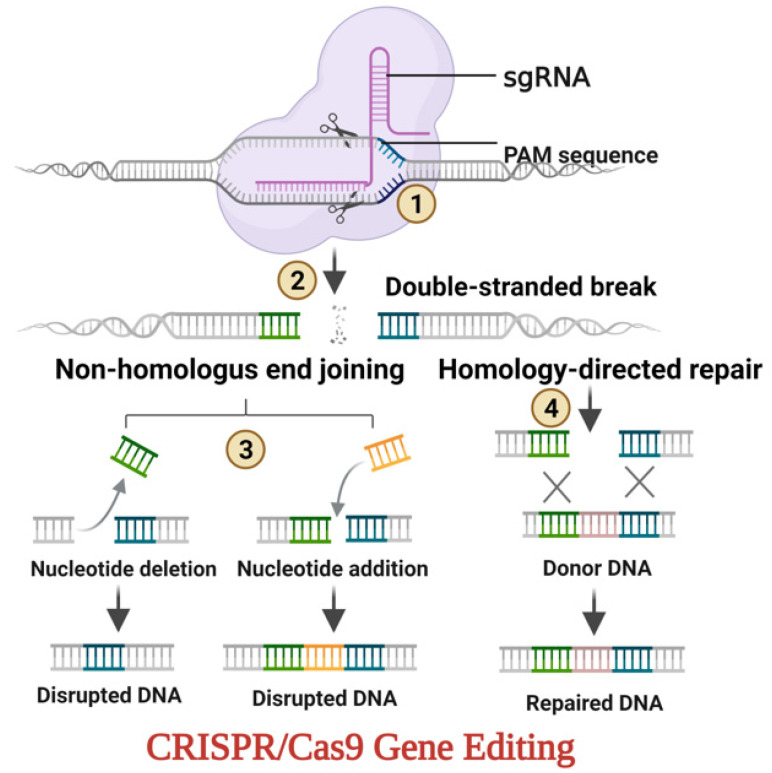
Overview of CRISPR-Cas9 mediated gene editing. ① A complex of CRISPR-Cas9 genetic scissors and artificially constructed single guide RNA (sgRNA) scans DNA and traces code where a cut has to be made. ② Formation of non-homologous end-joining (NHEJ) and homology direct repair (HDR) strands after DNA double-strand break (DSB). ③ Ligation of the DNA DSB by nucleotides addition on the right and deletion on the left result in gene disruption. ④ Repairing the DSB in HDR by employing an externally provided homologous DNA template for copying. The donor template’s DNA sequence is duplicated at the targeted site, which results in a guided repair.

**Table 1 ijms-23-09982-t001:** QTL for potato tuber quality and quantity traits.

**Trait**	**QTL**	**Chromosome Number**	**References**
Starch granule size	SGS02-8 and SGS03-8	VIII	[16]
Starch contents	pPt-535988–pPt-538127, toPt-440651	I	[17]
	pPt-539763	II	
	toPt-437014–pPt-538033	III	
	toPt-438845	VIII	
	pPt-533878	X	
	pPt-471789	XI	
	pPt-656237	XII	
	STM1049-1, STWIN12G, STM1049-3	I	[18]
	EAAT_MCGA_381	III	
	EACG_MCAA_191 DS, STM1002-1, STM1002-2	IV	
	EACG_MCAT_925, StI022-3, StI022-5	VIII	
	EATC_MCCG_182	X	
	EACG_MCAA_191 DS, StI017-2DS	XI	
	EACC_MCGA_114, EACG_MCAA_119 DS	XII	
	PCT_MACT_86, StI022-2 DS, EACG_MCAA_119 DS	Unlinked	
	Pain1-A 718 (C 552) 2, Pain1- A 1544 and Pain1-T 741, Pain1-8c	III	[19]
	InvGE-A 85 (A 86), InvGE-G 95 (G 106)	IX	
	InvCD141_T 543 (A 280, T 288, T 339, A 630, C 1030, G 1031, T 1096), InvCD141-G 765	X	
	Stp23-8b	III	[20]
CIS	REC_B_05-1	V	[21]
	CIS_E_07-1	VII	
Tuber shape	Solcap_snp_c2_34875	IV	[22]
	Solcap_snp_c2_25485, Solcap_snp_c2_25510	X	
	Solcap_snp_c1_1847	I	[23]
	Solcap_snp_c2_54790	IV	
	Solcap_snp_c2_26012,	VII	
	Solcap_snp_c1_15594, Solcap_snp_c1_11535	X	
Tuber skin color	Solcap_snp_c2_31852, Solcap_snp_c2_25759, Solcap_snp_c2_21178	I	[22]
	solcap_snp_c1_12440	V	
	solcap_snp_c2_4342, solcap_snp_c2_45215	VII	
	solcap_snp_c2_50702, solcap_snp_c2_53902, solcap_snp_c2_15803	VIII	
	solcap_snp_c2_22697	X	
	solcap_snp_c2_39889	XI	
	and solcap_snp_c2_5385	XII	

**Table 2 ijms-23-09982-t002:** Comparison of three different kinds of transgenics to improve potato tubers’ quality and quantity.

**Transgenics**	**Trait Introduced/Modified**	**Gene Added/Silenced**	**References**
Conventional	Starch yield	*PsGPT*	[63]
	Tuber yield	*AtPAP2*	[64]
		*SPS*	[65]
		*tms1*	[66]
	Amino acid (methionine)	*AmA1*	[67]
	Amino acid (cysteine)	*cysE*	[68]
	Astaxanthin	*Or*	[69]
	Ketocarotenoids	*crtO*	[70]
	Carotenoids	*crtZ*,	[71]
		*bkt1*	[72]
		*StLCYb*	[73]
	Ascorbic acid	DHAR	[74]
	Vitamin-E	*At*-*HPPD* & *At*-*HPT*	[75]
	Vitamin B6	*PDXII*	[76]
	CIS	*SbRFP1*	[77]
		*NtInvInh2*, *StInvInh2A* and *StInvInh2B*	[78,79]
		*LbPFK*	[80]
RNAi	Starch quality	*SSIII*	[81]
		*GBSS*	[82]
	CIS	*VInv*,	[83,84,85]
		*SPP*	[86]
TALENS	Starch quality	*GBSS*	[87]
	CIS	*StvacINV2*	[88]
		*VInv*	[89]
CRISPR	Starch quality	*GBSS*	[90]
	Anti-browning	*StPPO2*	[91]
	Reducing steroidal alkaloids	*St16DOX*	[92]

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
