# Peer review of "Recent Advances in Molecular Improvement for Potato Tuber Traits"

_ijms, 2022, doi:10.3390/ijms23179982_

Round 1
Reviewer 1 Report
Authors in their review summarize recent methods of breeding for better potato tuber traits. They comprehensively and thoroughly review recent literature, current methods and defined molecular markers necessary for quick breeding.
Major point:
The main shortage of the presented article is English. Text is very hard to follow, with frequent grammatical errors and typos. Some text passages loosely follow, sometimes giving impression of simple “list of items”. The text needs principal language correction.
Minor points:
-section 2 “QTL Mapping and Molecular breeding” deals with linkage markers for various traits. There is not much information about breeding itself – I would thus avoid word “breeding” in headings and subheadings in this section.
-avoid using “viz.” in the text
-quality of figures is poor. Text looks blurred – it needs better resolution.
-table 2: check lines separating rows (individual “sections”).
Author Response
Reviewer 1
The main shortage of the presented article is English. Text is very hard to follow, with frequent grammatical errors and typos. Some text passages loosely follow, sometimes giving impression of simple “list of items”. The text needs principal language correction.
Reply: Thanks for pointing out the shortage of ms, we have carefully checked and corrected the English grammars.
-section 2 “QTL Mapping and Molecular breeding” deals with linkage markers for various traits. There is not much information about breeding itself – I would thus avoid word “breeding” in headings and subheadings in this section.
Reply: Yes, we agreed with your comment, and have removed the words of Molecular breeding.
-avoid using “viz.” in the text
Reply: corrected.
-quality of figures is poor. Text looks blurred – it needs better resolution.
Reply: The resolution of Figures have been improved.
-table 2: check lines separating rows (individual “sections”)
Reply: corrected.
Reviewer 2 Report
The review article by Ahmad et al has gathered updated information on recent advances in potato improvement through modern techniques. The style of presentation about the advances is mixed up in certain sub-sections which are indicated below.
1. Consider revising the title of the article.
2. is potato the 3rd important crop than maize/corn? It is in the 2nd line under introduction section.
3. Make the advances in genetics and breeding in separate headings. Under genetics part, place the important reported QTL in tabular form.
4. Similarly, for breeding/improvement, place the i.conventional breeding explaining the difficulties faced ii.molecular breeding approaches with good examples. Under this both Maker-assisted breeding and genomic-assisted breeding should be placed in this part iii. Transgenic approach iv. CRISPER etc. Please do not mix conventional with the molecular breeding approach.
5. For the genomic-assisted breeding sub-section, try to give a model breeding flow diagram for ease of understanding. The various examples on genetic gain of Denmark and other places should beautifully be arranged without mixing all and making like a note. Also, indicate about the best models for estimation of markers effects.

Author Response
The review article by Ahmad et al has gathered updated information on recent advances in potato improvement through modern techniques. The style of presentation about the advances is mixed up in certain sub-sections which are indicated below.
- Consider revising the title of the article.
Reply: We think the title is suitable. We basically focused on the tuber quality and quantity (yield).
- is potato the 3rd important crop than maize/corn? It is in the 2nd line under introduction section.
Reply: we corrected it as “Potato (Solanum tuberosum L.) is the most important non-grain food crop concerning its global consumption”.
- Make the advances in genetics and breeding in separate headings. Under genetics part, place the important reported QTL in tabular form.
Reply: Since most QTL mapping work belongs to Genetics, and few breeding work done before, we have removed the Molecular breeding in the section title as suggested by Reviewer 1.
- Similarly, for breeding/improvement, place the i.conventional breeding explaining the difficulties faced ii.molecular breeding approaches with good examples. Under this both Maker-assisted breeding and genomic-assisted breeding should be placed in this part iii. Transgenic approach iv. CRISPER etc. Please do not mix conventional with the molecular breeding approach.
Reply: As stated in point 3, we deleted the words of molecular breeding, since there are no good examples of molecular breeding work. We have already provided the Crispr work with different subtitles from others (4.3.2 CRISPR-Cas mediated potato modifications ).
- For the genomic-assisted breeding sub-section, try to give a model breeding flow diagram for ease of understanding. The various examples on genetic gain of Denmark and other places should beautifully be arranged without mixing all and making like a note. Also, indicate about the best models for estimation of markers effect.
Reply: Thanks for the good comments, we have added a new figure (Figure 1) providing a schematic flow diagram of genome-assisted breeding. Results of some best models have provided.
Reviewer 3 Report
Review article entitled Recent advances in molecular improvement for potato tuber traits has been well compiled, all the required information has been mentioned with recent references. Figure and tables presented are also fine. Although manuscript requires minor improvement as mentioned below.
Introduction
First paragraph required English improvements.
Paragraph 3rd meaning of first sentence is not clear.
Last paragraph requires extensive language editing
QTL Mapping and Molecular Breeding
Re-write entire paragraph
2.3. Breeding to improve Tuber Morphology
Last paragraph is not clear
3. Genome Selection
Entire section requires language improvement
Figure 1: Improves the figure quality
Figure 2: Figure can be presented in more better way by modifying the pictures
Table 2: Some words requires abbreviation
Author Response
Review article entitled Recent advances in molecular improvement for potato tuber traits has been well compiled, all the required information has been mentioned with recent references. Figure and tables presented are also fine. Although manuscript requires minor improvement as mentioned below.
Introduction
First paragraph required English improvements.
Reply: Thanks for the comment, we have checked and corrected all the errors in English grammars.
Paragraph 3rd meaning of first sentence is not clear.
Reply: We have corrected it to “The main motive for improving this crop is to improve the processing tuber qual-ity with a higher yield.”.
Last paragraph requires extensive language editing
Reply: We have thoroughly re-written the last paragraph.
QTL Mapping and Molecular Breeding
Re-write entire paragraph
Reply: We have thoroughly re-written the section 2. Quantitative trait loci (QTL) mapping.
2.3. Breeding to improve Tuber Morphology
Last paragraph is not clear
Reply: The last paragraph has been re-written carefully.
- Genome Selection
Entire section requires language improvement
Reply: we have checked and corrected all the errors in English grammars.
Figure 1: Improves the figure quality
Figure 2: Figure can be presented in more better way by modifying the pictures
Reply: The resolution and quality of Fig 1 & 2 (now Figure 2 &3) have been improved.
Table 2: Some words requires abbreviation
Reply: The CIS (cold‐induced sweetening) is a common name widely used in this ms, the others are gene names, and most of which have been explained in the text.
Round 2
Reviewer 1 Report
Text has been thoroughly revised and significantly improved. Now the article is much easier to read and follow.
All my points were addressed, so I do not have any other major objections.
Some minor points/suggestions:
-Fig. 1 is located before section 3, but its first citation is already mentioned 3 pages earlier (line 216) - move it closer to the Table 1?
-line 21: "literature"
-line 200: "interesting traits"
-line 201: "tetrasomic polypoidity"
-line 499: "important quality trait"
-lines 528-531: I would insert a citation after "2016", I would also change "to be significant in 2017" to "later" to avoid misunderstanding that this locus was significant only in the year 2017.
-line 535: "QTLs"
-line 800: "typical selection"
-line 802: "high-yield"
-line 991: "will be discussed here"
-line 1461: "genetically modified"
-line 1523: "have been reported" seems to me redundant - either delete or rephrase the sentence
Author Response
-Fig. 1 is located before section 3, but its first citation is already mentioned 3 pages earlier (line 216) - move it closer to the Table 1?
Response: we have moved Fig. 1 to the page 3 now.
-line 21: "literature"
Response: changed.
-line 200: "interesting traits"
Response: changed, in Line 46.
-line 201: "tetrasomic polypoidity"
Response: changed, in Line 47.
-line 499: "important quality trait"
Response: the word quality was added in Line 153.
-lines 528-531: I would insert a citation after "2016", I would also change "to be significant in 2017" to "later" to avoid misunderstanding that this locus was significant only in the year 2017.
Response: a citation was added after 2016. This study were conducted in 2016 and 2017, two years, so one locus is found in 2016, while another was in 2017. Now in Line 217-219.
-line 535: "QTLs"
Response: changed.
-line 800: "typical selection"
Response: changed in Line 188
-line 802: "high-yield"
Response: changed in Line 190
-line 991: "will be discussed here"
Response: changed. In Line 248.
-line 1461: "genetically modified"
Response: changed. In line 320.
-line 1523: "have been reported" seems to me redundant - either delete or rephrase the sentence
Response: we have changed it to “have been identified” in Line 330.